# RUDE-AL: Roped UGV Deployment Algorithm of an MCDPR for Sinkhole Exploration

**DOI:** 10.3390/s23146487

**Published:** 2023-07-18

**Authors:** David Orbea, Christyan Cruz Ulloa, Jaime Del Cerro, Antonio Barrientos

**Affiliations:** Centro de Automática y Robótica (UPM-CSIC), Universidad Politécnica de Madrid—Consejo Superior de Investigaciones Científicas, 28006 Madrid, Spain; christyan.cruz.ulloa@upm.es (C.C.U.); j.cerro@upm.es (J.D.C.)

**Keywords:** genetic algorithms, multi-robot system, CDPR, MCDPR, ROS, roped fleet, navigation

## Abstract

The presence of sinkholes has been widely studied due to their potential risk to infrastructure and to the lives of inhabitants and rescuers in urban disaster areas, which is generally addressed in geotechnics and geophysics. In recent years, robotics has gained importance for the inspection and assessment of areas of potential risk for sinkhole formation, as well as for environmental exploration and post-disaster assistance. From the mobile robotics approach, this paper proposes RUDE-AL (Roped UGV DEployment ALgorithm), a methodology for deploying a Mobile Cable-Driven Parallel Robot (MCDPR) composed of four mobile robots and a cable-driven parallel robot (CDPR) for sinkhole exploration tasks and assistance to potential trapped victims. The deployment of the fleet is organized with node-edge formation during the mission’s first stage, positioning itself around the area of interest and acting as anchors for the subsequent release of the cable robot. One of the relevant issues considered in this work is the selection of target points for mobile robots (anchors) considering the constraints of a roped fleet, avoiding the collision of the cables with positive obstacles through a fitting function that maximizes the area covered of the zone to explore and minimizes the cost of the route distance performed by the fleet using genetic algorithms, generating feasible target routes for each mobile robot with a configurable balance between the parameters of the fitness function. The main results show a robust method whose adjustment function is affected by the number of positive obstacles near the area of interest and the shape characteristics of the sinkhole.

## 1. Introduction

Sinkholes are hollows in the ground surface formed by the dissolution of limestone or geological features, considered closed cavities drained due to subsoil dissolution in karst rocks [1]. This land subsidence and collapse represents a major hazard that substantially impacts economic and human losses [2], as well as in nearby infrastructures [3]. In the last decade, several karst collapses have taken place, causing the collapse of roads and buildings in urban areas, causing safety risks to residents [4]. Different natural storms have caused sinkholes such as in Guatemala (Guatemala City—2010) [5], USA (Florida—2004) [6].

Pressure wireless sensor network (WSN) technologies and neural network learning databases use UAVs and thermal cameras for sinkhole detection and monitoring [7]. There are also approaches for the use of robotic total stations (RTS) together with total stations (TS) to calculate the horizontal and vertical displacement of the earth for sinkhole detection [8], opening the line of integration of mobile robots and their application for monitoring, exploration, and assistance for search and rescue (SAR) tasks.

The success of search and rescue (SAR) missions depends on the performance of robotic platforms individually [9]. However, search and exploration tasks can be enhanced through cooperative systems using multiple unmanned ground vehicles (UGVs) [10]. There are several types of cooperative systems used in urban search and rescue (USAR) [11], wilderness search and rescue (WiSAR) [12] and air-sea rescue (ASR) operations [13].

Regarding deployment of robot fleets in disaster environments, [14] details of thirty-four official deployments have been reported and analyzed, where UGVs have been used in twenty-two incidents. In contrast, UAVs and unmanned aquatic vehicles (UMVs) in eight incidents [15], mainly at the subway level in mining environments, building collapse, earthquakes, and hurricanes, due to their potential to save lives by offering faster response times [16], support in hazardous environments [17], real-time monitoring [18], and area mapping [19].

The most important robotic challenges in subway environments are the limitation of wireless communications and detection task such as toxic gas analysis. Robots used for WiSAR tasks must be prepared for difficult weather conditions and interfere minimally with environmental data acquisition [20], so robots with mobile base cables with robust situational awareness are currently proposed for autonomous navigation, where environmental monitoring sensors are mounted for gas discrimination [21], air quality sensing, or gas concentration [22], with which information assistance is provided to rescue personnel for hazardous area detection.

In terms of SAR tasks using mobile robots, these systems can be considered cooperative Cable-Driven Parallel Robots (CDPR), where each robot represents an anchor point for the wires. This configuration is a major challenge for base localization in the environment, obstacle avoidance, and adaptive control of payload position and orientation [23].

This work is part of a Robotics and Cybernetics Group (ROBCIB) of the Polytechnic University of Madrid (UPM) project, which main objective is to perform the deployment of a Mobile Cable-Driven Parallel Robot (MCDPR) inside a sinkhole automatically, considering mobile robots as mobile bases that will release a parallel cable-driven robot in the region of interest. Therefore, challenges are related to the deployment of the mobile robots (mobile bases) and relative localization of the CDPR. Accordingly, the strategy is focused on two stages: first, the traversability of a path for the mobile bases from the base station to the surroundings of the sinkhole, considering that the mobile bases of the MCDPR will be attached by ropes; then, the release of the CDPR, avoiding contact with positive obstacles. Regarding the relative localization challenge, in order to get a good precision of portable localization systems (radar, infrared, UWB), using trilateration methods, a first approach presents the use of a fleet of four mobile robots and a CDPR carried on top of one of the mobile platforms, applying two physical configurations: node-edge fleet and roped individual.

This paper’s main contribution is RUDE-AL (Roped UGV Deployment Algorithm), an algorithm for the autonomous selection of anchoring points of a mobile robot fleet for sinkhole exploration through computer vision techniques to filter positive and negative obstacles, and genetic algorithms for solving optimization problems.

The mission comprises two stages: the first one consists of finding a feasible fleet trajectory in a node-edge configuration with four nodes and three edges; the second corresponds to the deployment of each robot to its corresponding vertex.

For this purpose, multiple tests were carried out on 2D images representing 2.5D maps, which were generated manually (using image edition tools) and automatically (images containing splines generated using Bézier curves), evaluating the behavior of the algorithm to produce feasible roped navigation paths for environments with different physical characteristics.

Implementing a heuristic method (genetic algorithms) considerably reduces computational time, allowing to adjust the weights of the variables to be optimized through a weighted fitness function for multi-objective optimization. The algorithm is applied to the navigation of a fleet of four simulated robots in Gazebo.

The work is made up of the following sections. Section 2 presents the state of the art and related works. Section 3 details the problem formulation and the proposal. Section 4 describes the generation of the different environments for testing and description of the performed experiments. Section 5 shows the results obtained by the target point selection algorithm. Finally, Section 6 presents the conclusions obtained.

## 2. Related Work

### 2.1. Mobile CDPRs

CDPRs are emerging as an attractive proposition due to their advantages in covering large volumes with fast movements while maintaining a balance between light weight and high durability [24].

CDPR-related research has been reported since 1984, originally for underwater applications. The RoboCrane project [25], developed under the Defense Advanced Research Project Agency (DARPA), extended CDPR applications to land, sea, air, and space. CDPRs show high load-to-weight ratio, while relying on stable configurations, flexibility, and maneuverability over rough terrain surfaces.

An example of mobile CDPRs are the so-called extended-crane systems, which represent a combination between a cable robot and a conventional crane, where the robotic configuration is intended to separate positioning and balancing tasks. Another example of a reconfigurable structure for cable robots is found in agricultural applications, adding mobile pillars to transport winches while controlling the position of the mobile platform.

In terms of CDPR applications, the IPAnema family of robots [26] is used for industrial inspection, handling, and assembly tasks, heavy lifting, CoGiro [27], motion simulators, CableRobot [28], and recreation of underwater environments [29]; in logistics warehousing tasks, CABLAR [30] and FASTKIT [31], and in search and rescue operations, the MARIONET family of robots [32].

At conceptual level, the MoPick prototype [33] presents an approach to parallel cable robots with mobile bases (MCDPR) for pick and place tasks. A parallel cable robot for multiple mobile cranes (CPRMCs) is proposed by [34], detailing the design and a multilateration-based localization algorithm, whose global planning is performed with a grid-based artificial potential field method. It also uses sensors for cooperative obstacle avoidance, integrating autonomous level control for the platform, together with a co-simulation by using Matlab and Labview.

A CDPR with three mobile cranes for search and rescue operations is shown by [35], where each mobile base consists of a reconfigurable telescopic boom that can rotate (Figure 1a). The cable is mounted from the tip of the telescopic arm to the end effector. Although the cranes are fixed, the system can be reconfigured to increase the working space and keep the system in static equilibrium. The objective of this study is to compare the stress distribution and the size of the working space when applying payloads.

Considering strategies for motion planning of a fleet of mobile robots for deployment of a cable robot, ref. [36] proposes a modular CDPR carried by a rover for inspection and light manipulation tasks on celestial bodies (Figure 1b), whose applications focus on solar panel inspection and maintenance, as well as lava cave exploration (Figure 2).

In the related works described in Table 1, no strategies for the implementation of a self-contained portable CDPR have been found.

### 2.2. Multi-Objective Optimization

State-of-the-art path planning methods are commonly formulated based on a single-objective optimization for distance cost [37]. However, more factors need to be considered in real-world applications, which turns the study problem into a multi-objective optimization method applicable for multi-robot fleet path planning [38]. Common approaches use methods such as summing the weights of each objective as a function, known as the scalarized approach or weighted sum method [39], whose importance lies in correctly deciding the weight coefficients based on empirical checks.

The use of multi-objective genetic algorithms has been widely studied in building design [40], balancing of operations in hydroelectric reservoirs [41], sizing of microgrid systems [42], and sustainable mechanization allocation for spraying and harvesting systems [43]. It has been shown relevant in robotics for trajectory optimization [44,45], controller design [46], industrial robotic arm design [47], and cloud robotic platform service scheduling [48], and for multi-robot trajectory planning for area coverage [49].

On the application of evolutionary algorithms (EA) for target selection in multi-robot systems, ref. [50] proposes the use of an evolutionary algorithm with Indirect Representation and Extended Nearest Neighbor (IREANN) with a simple mutation operator for GTSPC (Generalized Travelling Salesman Problem with Coverage). The trajectory optimization for MDCPR of the MoPick platform using direct transcription optimization method where the optimization task adds the CDPR constraints in planning, and direct transcription increases the confidence of the data, is studied by [51].

## 3. Methodology

### 3.1. Problem Statement

As explained in Section 1, the strategy proposed to perform the deployment of the MCDPR must assure the traversability of a path for the mobile bases from a start point to the surrounds of the sinkhole, considering that the robots are attached by ropes that cannot pass through positive obstacles. After that, using optimization techniques, a feasible configuration that minimizes the fleet path distance cost and maximizes the area of the region of interest (ROI) to be covered must be found.

For the conceptualization of the problem, the starting point is a 2D map of free space, positive and negative obstacles. Figure 3a shows positive obstacles (gray) which block the traversability of mobile robots and ropes, while negative obstacles (black) block the traversability of mobile robots, but allow the passage of ropes. The figure is the actual image used as an input for the proposed algorithm.

Figure 3b shows a Gazebo representation of a structured environment, where the undercut corresponds to a negative obstacle.

Physically, the self-contained CDPR is on top of one of the mobile robots (Figure 4). However, the scope of the proposed algorithm is in two dimensions, so the corded fleet as seen from the top plane is four robots with one node from which the cables are projected.

Before performing path planning, the possibility of finding nearby positive obstacles that affect feasible target points for fleet route planning or deployment route must be considered. For these cases, a range of up to two nearby positive obstacles affecting the planning is defined, dividing into zones of interest where the user selects which area to  explore.

Once the feasible candidates are available, the fleet route planning is performed for each candidate point, calculating the distance cost of each route. Then, candidate point ranges are defined for each mobile robot, where for each combination of candidate points the area covered is calculated. The distance cost of the minimum fleet route is minimized and the area covered is maximized for each combination of candidate points using a weighted fit function.

Through the optimization of the fitness function, the target points are obtained, with which the deployment routes between them are planned (Figure 5a); subsequently, the feasible deployment routes are evaluated, avoiding collisions of the ropes with positive obstacles near the sinkhole to be explored (Figure 5b). Finally, the mission is executed in the simulator, adding a 2D display of the roped fleet.

### 3.2. RUDE-AL Algorithm

The main contribution of this article is the Roped UGV fleet DEployment ALgorithm (RUDE-AL), designed to provide a feasible goal selection based on the minimization of distance cost of fleet deployment and maximization of covered area, considering rope contact restrictions for negative and positive obstacles. It has been divided into two phases and three stages:Phase 1. Node-edge fleet configuration-Map, obstacle, and start selection for offline planning-Definition, sorting, and choosing of feasible goal candidates based on area coverage and distance cost of fleet deploymentPhase 2. Roped individual configuration-Goal local planning and establishment of deployment configuration based on rope restrictions

The flow diagram of the process of RUDE-AL is shown in Figure 6.

#### 3.2.1. Phase 1. Node-Edge Fleet Configuration

In this phase, the four robots are formed in a worm configuration consisting of four nodes and three edges (Figure 7). For navigation, the strategy used is to assign the front robot as the leader, and the others as followers. Each robot is assigned the fleet route, and minimum distance constraints are added to avoid collisions between them.

##### Map, Obstacle, and Start Selection for Offline Planning

In this stage, computer vision techniques are used to generate two-dimensional maps of positive and negative obstacles, separating them into individual masks for individual processing. Table 2 shows the associated color coding, as well as land navigation conditions and roped fleet restrictions for traversability.

OpenCV tools are used for contour detection, obstacle inflation (erode and dilate tools), and graphical representation of the algorithm. The starting point of the mission and the negative obstacle to be explored are manually selected. The existence of positive obstacles near the undercut that affect the roped deployment process is evaluated, and a group of candidate points is automatically obtained. Figure 8 shows the candidate point selection process for maps with several positive obstacles and one negative obstacle.

Figure 9 shows the process for selecting candidate points with a positive obstacle close to the zone of interest.

Figure 10 shows the selection of candidate points when there are two positive obstacles close to the area of interest.

Then, path planning is performed with a multi-query planner, in this case, a probabilistic roadmap (PRM), in order to obtain the candidate fleet routes from the starting point to the candidate target points of the first stage (Figure 11).

Algorithm 1 shows the pseudocode implemented for this stage, using as input variables: the required map, which is an RGB image in png format with a resolution of 1000 × 1000 pixels, whose colors correspond to the requirements of Table 2, and the value of the starting point start. The following functions explain the processes:Erode (line 12), Dilate (line 13), Range (line 14), Contours (line 16) and Centroids (line 18): are OpenCV functions used to inflate regions of images, define masks, and get characteristics of contours.Remove_dup (line 28): is a function to delete the multiple occurrences of an object in a list.Line(line 35): is a function that returns the slope and the constant “b” for a y = mx + b line between two input pointsCheck_click (line 51): is a function that determines where the user clicks and returns the selected zone to explore (options are 1 or 2).Route (line 51) is a function that performs the prm path planning between two points. Returns the feasible path between them.
**Algorithm 1** Fleet Planning  **Input:** map,start  **Output:** $c_{point},c_{path}$1:*//init_candidates: set of initial candidate points*2:*//$c_{point}$: set of candidate points*3:*//$c_{path}$: set of candidate paths*4:*//index_roi: index of the negative obstacle to explore*5:*//index_pos_obst: list of index of positive obstacles close to ROI*6:*//m: slope of line between two points*7:*//b: constant of line between two points*8:*//y_sup: bool that defines if “y” coordinate of a point is greater that the one described by equation constants*9:*//upper_c: candidate points greater that line defined between two near positive obstacles*10:*//lower_c: candidate points lower that line defined between two near positive obstacles*11:map←gray(map)12:erodedn_m←Erode(map)13:erodedp_m←Dilate(map)14:black_m←Range(erodedn_m,0,10)15:gray_m←Range(erodedp_m,50,150)16:neg_contours←Contours(black_m)17:pos_contours←Contours(gray_m)18:pos_centroids←Centroids(pos_contours)19:$c_{point}\leftarrow neg\_contours\left[index\_roi\right]$20:**for** j∈pos_centroids **do**21:    **for** $j\in neg_{c}ontours$ **do**22:        **if** gray_m[c_point[i]]==1 **then**23:           Delete from c_point this neg_contours [i]24:           Append in index_pos_obst this j25:        **end if**26:    **end for**27:**end for**28:index_pos_obst←Remove_dup[index_pos_obst]29:**if** length(index_pos_obst)==1 **then**30:    **for** $i\in c_{point}$ **do**31:        $c_{path}=Route\left(start,i\right)$32:    **end for**33:**end if**34:**if** length(index_pos_obst)==2 **then**35:    m,b←Line(pos_centroid[index_pos_obst[0]],pos_centroid[index_pos_obst[1]])36:    **for** $i\in c_{point}$ **do**37:        $y_{sup}\leftarrow check\_line\left(c_{point},m,b\right)$38:        **if** y_sup==true **then**39:           Append in upper_c this $c_{point}\left[i\right]$40:        **else**41:           Append in lower_c this $c_{point}\left[i\right]$42:        **end if**43:    **end for**44:    zone=Check_click()45:    **if** zone==1 **then**46:        $c_{point}=upper\_c$47:    **else**48:        $c_{point}=lower\_c$49:    **end if**50:    **for** $i\in c_{point}$ **do**51:        $c_{path}\left[i\right]=Route\left(start,i\right)$52:    **end for**53:**end if**54:**return**$\;c_{point},c_{path}$

##### Definition, Sorting, and Choice of Feasible Goal Candidates Based on Area Coverage and Distance Cost of Fleet Deployment

At this stage, the distance cost of the fleet route is minimized and the area covered by the polygon formed by the combination of four points corresponding to the position of the mobile robots is maximized. The candidate points are defined as $c_{1}$, $c_{2}$, $c_{3}$, $c_{4}$. Initially, to reduce the number of combinations to be processed, a minimum distance criterion is applied between subsequent points, defining minimum and maximum indices for each point in the general list of candidates, as shown in Figure 12, from which different vectors are obtained for each candidate point.

The calculation of the area covered by the combination of the candidate points is made with the sum of the areas of the triangles that form the quadrilateral 1-2-3-4, shown in Figure 13, with the Equation (Equation 1):(1)$$a_{cov}=\frac{1}{2}|\vec{a}\times \vec{b}|+\frac{1}{2}\left|\vec{b}\times \vec{c}\right|$$
where:$a_{cov}$: covered area by points$\vec{a}$: vector between points 1–4$\vec{b}$: vector between points 1–3$\vec{c}$: vector between points 1–2

For the calculation of the distance cost, in each fleet route associated to each candidate point (Figure 11), the Equation (Equation 2) is used:(2)$$c_{route}=\sum_{i=1}^{i=n}d\left(p_{i},p_{i-1}\right)$$
where:$c_{route}$: accumulated distance cost of route*d*: euclidean distance between points

The cost values associated to each candidate route are normalized with respect to the maximum distance cost. Since the aim is to maximize the fitness function, and the area covered calculated with respect to the area of the obstacle obtained with OpenCV, a weighted fitness function is defined using the scalarized approach (weighted sum method) with the Equation (Equation 3):(3)$$f_{fitness}=\frac{w_{area}*a_{candidate}}{a_{obstacle}}+w_{route}*max\left(inv\_route\_cost_{candidates}\right)$$
where:$f_{fitness}$: fitness function$w_{area}$: weight applied for area coverage$a_{candidate}$: weight applied for path distance cost$a_{obstacle}$: covered by 4-candidate points combination$w_{route}$: area of obstacle obtained from contours$inv\_route\_cost_{candidates}$: vector of inverse normalized route cost for candidate points

To estimate candidates that maximize the fitness function while reducing the computational cost, genetic algorithms, implemented with the PyGAD package [52], are used. To use the package, it is necessary to adapt the problem to the structure of a genetic algorithm.

A combination of four candidate points is sought, whose coordinates are defined with values of type “integer”, where each candidate point is contained in a pool of feasible solutions, and has an associated fleet path cost value. For the application of PyGAD to the problem, the indices of the lists of each candidate point are used as solutions, having as objective the maximization of the $f_{fitness}$ function, delimiting the maximum values of mutation range to the length of the vector of each feasible candidate. The data type is of type “int”. For the generation of the instance, the parameters described in Table 3 are set.

The value of $w_{area}$ is 2, and $w_{route}$ is 3, prioritizing the minimization of the fleet route distance cost, which represents the process with the highest energy cost due to the relationship with the number of robots in the fleet.

Algorithm 2 takes as input variables the candidate points $c_{point}$, and the paths of each candidate $c_{path}$. The following functions explain the process:Get_area (line 13) is a function that gets the area of the contour made by a list of points. The output is the area of the contour.Route_cost (line 14) is a function that calculates the accumulated individual distance cost for a list of paths.Index_range (line 15) is a function that defines the lower and upper limit indexes for each candidate point according to a minimum distance between points. The output is a range of index for each of the four candidate points.Get_candidates (line 16) is a function that defines independent lists of candidate points of a main list according to the given limits. The output is four lists of points.Fitness_function (line 17) is a function that iterates testing different combinations according to Ga_instance parameters. It includes the Area_calc() in order to calculate area for every iteration different combination of points.Area_calc (line 17) is a function that calculates the area described between four points according to cross product and sum of areas described in Equation (1).Ga_instance (line 18) is pygad instance to configure the genetic algorithm that include the parameters described in Table 2. The instance must run using Ga_instance.run. The output of Ga_instance.best_solution() is the best combination of four points according to the fitness function.
**Algorithm 2** Points Selection  **Input:** $c_{point},c_{path}$  **Output:** sol_points,fleet_path1:*//roi_area: area on negative obstacle in $pixels^{2}$*2:*//$c_{point}$: set of candidate points*3:*//$c_{path}$: set of candidate paths*4:*//range_c_points: 2x4 matrix of range of points for candidates 1, 2, 3, 4*5:*//c1: list of candidate 1 points*6:*//c2: list of candidate 2 points*7:*//c3: list of candidate 3 points*8:*//c4: list of candidate 4 points*9:*//sol_points: solution of four points*10:*//fit_function: fitness function used in pygad instance*11:*//sol_paths: fleet paths for solution points*12:*//path_cost: list of distance costs related to the input $c_{path}$*13:$roi\_area\leftarrow Get\_area(c_{point})$14:path_cost=Route_cost($c_{path}$)15:range_c1, range_c2, range_c3, range_c4=Index_range($c_{point}$)16:c1, c2, c3, c4=Get_candidates(range_c_points, $c_{point}$)17:fit_function=fitness_function(c1,c2,c3,c4, roi_area, path_cost, Area_calc())18:ga_instance(fit_function, range_c_points)19:sol_points=ga_instance.best_solution()20:**for** i∈sol_points **do**21:    sol_paths[i]=$c_{path}$[sol_points[i]]22:**end for**23:fleet_path=min(sol_paths)24:**return** sol_points,fleet_path

#### 3.2.2. Phase 2. Roped Individual Configuration

In this phase, the robots are freed from the constraints of the worm configuration, where each mobile robot is attached by a rope to the parallel cable robot (Figure 14). In this phase, it is important to note that the lead robot will not always carry the wire robot, as the robot to carry will be defined by obtaining the feasible deployment configuration. Each robot in the fleet will have its own route, added to the fleet route from the previous phase.

##### Goal Local Planning and Establishment of Deployment Configuration Based on Rope Restrictions

After having selected the best feasible candidate points, a route planning is performed between the selected points to obtain the deployment routes and propose a feasible deployment configuration considering the roped fleet constraints (Table 2). By having one fleet route and several deployment routes, different combinations of possible routes can be generated, shown in Figure 15.

The point corresponding to the fleet robot carrying the cable robot is also taken into account since, in addition to generating a feasible deployment path for UGV navigation, corded fleet constraints for positive obstacles must be considered; see Figure 16.

In addition to the previous considerations, it must be evaluated that at each instant of the deployment, there is a rope connecting each mobile robot with the cable robot, where contact with positive obstacles must be avoided; see Figure 17.

Table 4 details the conditions considered to obtain feasible paths and feasible configurations for possible situations using different start positions.

Finally, the routes of each robot are defined according to the possible configurations for fleet deployment (Figure 18), using the criteria defined in Table 5.

The criterion for selecting robot routes is to apply the longest route to the leader robot, progressively descending to the shortest route to follower robot 3, in type 1 and type 2 configurations. For type 3 and type 4 configurations, the path to the longest boundary node is assigned to the leader robot, the shortest leg of the path from the leader robot to follower robot 1, the path to the shortest boundary node to follower robot 2, and the shortest path to follower robot 3.

The pseudocode is detailed in Algorithm 3. The functions that help to understand this section are:Route (line: 10) is a function that performs the PRM path planning between two points. It returns the feasible path between them.Route_check (line: 14) is a function that creates an interpolated line between input point and every interpolated point of the input route, and checks if there are collisions between the interpolated line (rope) and a positive obstacle. Output is a Boolean true if there is collision, and false if there is no collisions.GetRobotPaths (line: 35) is a function in which input is feasibility of each path of the roped configuration deployment check, and it returns the path for each robot according to Table 5.
**Algorithm 3** Roped deployment  **Input:** sol_points,fleet_path,map  **Output:** fleet_path,r1_path,r2_path,r3_path,r4_path1:*//init_candidates: set of initial candidate points*2:*//check_path: is a Boolean that indicates if the path is feasible to navigate with roped fleet conditions (true=no valid, false=valid)*3:*//p1_check: is a list that contains boolean values related to a feasible path between point 1 and a route*4:*//p2_check: is a list that contains boolean values related to a feasible path between point 2 and a route*5:*//p3_check: is a list that contains boolean values related to a feasible path between point 3 and a route*6:*//p4_check: is a list that contains boolean values related to a feasible path between point 4 and a route*7:map←gray(map)8:gray_m←Range(map,50,150)9:**for** i∈(sol_points−1) **do**10:    deployment_paths[i]=Route(sol_points[i],sol_points[i+1])11:**end for**12:**for** i∈(sol_points) **do**13:    **for** $j\in deployment_{p}aths$ **do**14:        check_path=Route_check(sol_points[i],deployment_paths[j])15:        **if** check_path==true **then**16:           **if** i==1 **then**17:               Append in p1_check this check_path18:               **break**19:           **end if**20:           **if** i==2 **then**21:               Append in p2_check this check_path22:               **break**23:           **end if**24:           **if** i==3 **then**25:               Append in p3_check this check_path26:               **break**27:           **end if**28:           **if** i==4 **then**29:               Append in p4_check this check_path30:               **break**31:           **end if**32:        **end if**33:    **end for**34:**end for**35:robot1_path, robot2_path, robot3_path, robot4_path=GetRobotPaths(p1_check, p2_check, p3_check, p4_check)36:**return** robot1_path, robot2_path, robot3_path, robot4_path

## 4. Experiments

In order to test the performance of the algorithm on environments with different characteristics, two groups of environments are defined. The first, GLOBAL TEST, is intended to test different combinations of environments with various positive and negative obstacles, to test the robustness of the fleet in node-edge configuration; this group consists of manual generation of maps, using combinations associated with the characteristics of the fleet environment. The second group, SHAPE TEST, modifies the shape features of the region of interest to be explored, in order to test the robustness of the algorithm and identify the mission characteristics (shape of ROI, number of positive obstacles near the ROI, start point) that can affect the performance.

For GLOBAL TEST, the experiments are performed on six maps with the characteristics shown in Table 6.

For each map, feasible fleet paths are obtained just once to get the feasible fleet candidate path cost, because the main goal is evaluating the performance of algorithm, not to be a PRM path planner.

For GLOBAL TEST, the maps shown in Figure 19 are used with the characteristics described in Table 6.

For SHAPE TEST, thirty types of maps are automatically generated for each number of positive obstacles around the area of interest, varying the shape of the sinkhole. These maps are generated through a point connection script using cubic Bezier curves [53]. The criteria for autogeneration of the maps are explained in Table 7.

Where:Maps: The range of maps that are auto-generated with the indicated characteristics.$N_{random}$: number of random points to connect to generate the Bezier curve.$Cp_{rad}$: radius around the points where control points are. A larger radius means a sharper feature.Smoothness: Parameter to define the smoothness of the curve.Scale: X and Y pixel rectangle size where the random points will be generated.

The total number of generated maps is 90. Tests are performed by planning the routes from four start points ([100,100], [100,900], [800,100], [800,800]), with a total of 360 tests of the algorithm.

The 1000 × 1000 pixel resolution maps have a scale of 0.1 m per pixel, thus, the size range of the generated sinkholes goes from 20 m to 50 m approximately. The reference mobile robot size is the SummitXL (0.7 m length, 0.6 m width, 0.45 m height).

Some of the used maps are shown in Figure 20.

All experiments were carried on Intel Core i7-9750H PC with 16GB RAM, under the ROS Noetic operating system and Python 3.8, on a computer running Ubuntu 20.04. For simulation, Gazebo is used through Robotnik Stack for Summit-XL [54], using mobile platforms of four wheels with differential locomotion.

## 5. Results

This chapter presents the results of the experiments. Table 8 shows the summary of the results obtained, detailing the exceptions for SHAPE TEST, which are explained in the following sections. The results are encouraging, allowing us to identify the circumstances of the map that caused incorrect results.

Tests are performed from the starting point [100,100], obtaining a feasible fleet route, deployment routes, and cable layout. Figure 21, Figure 22, Figure 23, Figure 24, Figure 25 and Figure 26.

In GLOBAL TEST experiments, the algorithm finds feasible solutions for 100% of the cases.

In SHAPE TEST experiments, maps with the features in Table 7 are tested. Figure 27 shows some of the results obtained by the algorithm in the experiments performed in the SHAPE TEST environment.

For the 360 tests, the algorithm returns feasible routes in 357, representing 99.16% efficiency for fleet route and deployment route generation.

Since the scope of the paper is delimited to point selection and feasible route generation for a fleet of corded robots, the workspace of the effector mounted on one of the robots is not considered. However, post-fleet navigation issues are discussed, through caveats defined in two groups: limitations of the corded robot workspace, and risk of collision of the cords in the release of the end effector.

Selected points representing a limitation to the workspace are obtained eight times (Figure 28), i.e., 2.22%, while the risk of cable collision with positive obstacles occurs 10 times (Figure 29), 2.78% of the time.

There are also exceptional cases where navigation and deployment of the fleet is feasible, but the release of the effector would present an error. These cases occur in six occasions (1.67%), shown in Figure 30.

### 5.1. Fitness

In order to analyze statistical significance, box and whisker plots associated with the relationships between the fit variables (fitness, weighted fleet route distance cost, weighted area covered) and map characteristics (number of positive obstacles in the ROI, shape characteristics for map generation, starting points) are presented.

Figure 31 shows the box and whisker plots of fitness and map characteristics. The one-way analysis of variance (ANOVA) shows that there are significant differences with α = 0.05 of fitness vs. number of positive obstacles in the ROI (F = 305.53, *p* = 0), fitness vs. shape (F = 4.04, *p* = 0.0184), and that there are no significant differences of fitness vs. starting point data (F = 1.08, *p* = 0.3588).

Figure 32 collects the plots for the weighted cost of the area covered and the map features. Analysis of variance indicates significant differences with α = 0.05 of area weighted cost vs. number of positive obstacles in the ROI (F = 266.65, *p* = 0), area weighted cost vs. shape (F = 3.4, *p* = 0.0345), and that there are no significant differences of area weighted cost vs. start point data (F = 0.07, *p* = 0.9748).

Figure 33 points out the plots for the weighted fleet route distance cost and map features. Analysis of variance reveals significant differences with α = 0.05 of weighted fleet route distance cost vs. number of positive obstacles in the ROI (F = 9.64, *p* = 0), weighted fleet route distance cost vs. shape (F = 5.98, *p* = 0.0028), and that there are no significant differences of fleet route distance vs. start point data (F = 1.64, *p* = 0.1791).

In summary, it can be said that the fitness value obtained as a result of the algorithm is mainly affected by the number of positive obstacles close to the ROI, and secondarily, by the shape of the land depression to be explored. The starting point does not affect in a relevant way the fitness function value.

### 5.2. Algorithm Complexity

The complexity of an algorithm is usually calculated based on the Big-O notation, divided into time complexity, referring to the execution time of the algorithm, and space complexity, the amount of memory used by an algorithm [55]. In this section, the temporal complexity of the genetic algorithm is discussed. The amount of data processed by an algorithm is represented by N. If the algorithm does not depend on N, it has a constant complexity, represented by the notation O(1). If the algorithm depends on N, the complexity is represented as a function of this variable with the notations O(N), O($N^{2}$), O(log N), O(N log N), O($2^{N}$), O(N!). Figure 34 shows the evolution of the execution time of the genetic algorithm as a function of the number of samples. Input range goes from 76 to 806, with a maximum time of 16.55 s, and an average time of 9.48 s. A linear regression is performed to estimate the fit to the measured time data, obtaining the Equation (Equation 4):(4)$$time=0.0101*n_{inputs}+5.8094\;$$

With a value of $R^{2}$ of 0.5523, the algorithm has a proportional behaviour, understood in the Big-O notation as O(n), considered a fair complexity.

### 5.3. Test on Gazebo Simulator

For the fleet navigation simulation, the Summit XL robot ROS package is adapted to use four robots. The 3D environments are developed by approximating the splines of the generated maps and exporting to a compatible format for import into the Gazebo environment. A script is implemented to track the trajectories of each robot, and a viewer is set up parallel to the execution of the mission. Figure 35 shows the operation of the algorithm and the navigation of the robot fleet. The size scale used is 0.1 m per pixel for the map, so the sinkhole is approximately 50 m long, 30 m wide, and 20 m deep.

## 6. Conclusions

In this work, the use of a fleet of UGV land mobile robots is proposed for the deployment of a mobile cable-driven parallel robot to perform the exploration of sinkholes.

For the positioning of the mobile bases of the MCDPR, fleet route planning and corded deployment are performed using maps with different characteristics, generated either manually or automatically.

Several experiments have been performed by varying the fleet route environment and the features of the sinkhole to explore. The results of the experiments show the robustness of the algorithm for the generation of feasible routes, in addition to corroborating that both the shape of the sinkhole and the number of positive obstacles in the ROI region of interest present significant differences for the cost function fit. The validity of the routes in a simulation model in Gazebo has been also evaluated.

The use of evolutionary algorithms considerably reduces the calculation time of the algorithm. However, parameter settings should be tailored for each task, so it is important to initially consider the scope and scalability of the proposed solution.

Future work should focus on optimization of fleet routes, smoothing of planned routes to improve navigation, consideration of physical characteristics of ropes, and integration into real roped robotic fleets. For real-world scenarios, navigation can be improved with the addition of local planning techniques to avoid dynamic obstacles. Simulation can focus on testing the algorithm on rope capable simulators considering the dynamical restrictions of the real roped robotic fleet. In addition, the workspace of the released CDPR as part of the proposed weighted fitness function should be analyzed.

## Figures and Tables

**Figure 1 sensors-23-06487-f001:**
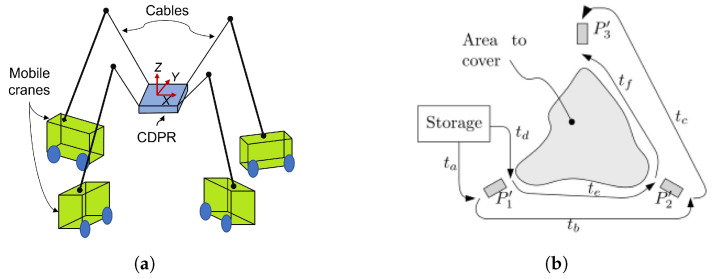
MCDPR system configurations and deployment. (**a**) Schematic of CDPR with mobile cranes. Source: Authors. (**b**) Deployment procedure of a rover. Obtained from [36].

**Figure 2 sensors-23-06487-f002:**
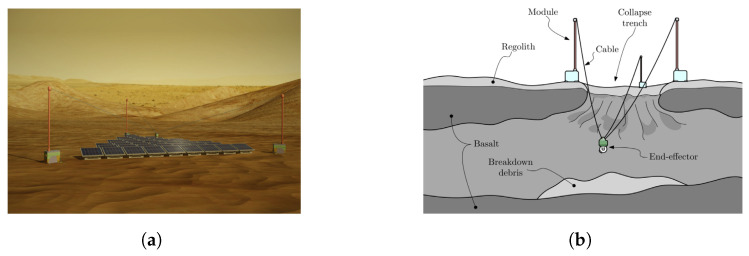
Applications of modular CDPR. (**a**) Maintenance of large ground solar array. Obtained from [36]. (**b**) Exploration of lava tubes on celestial bodies. Obtained from [36].

**Figure 3 sensors-23-06487-f003:**
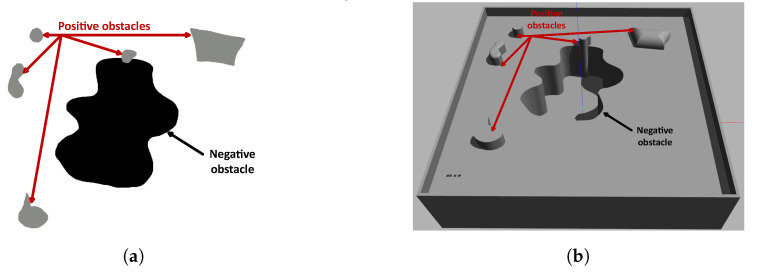
Map representation of environment. (**a**) A 2.5D map representation. (**b**) A 3D map representation. Source: Authors.

**Figure 4 sensors-23-06487-f004:**
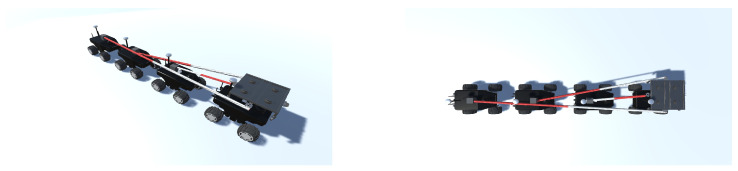
Physical configuration of the fleet. Source: Authors.

**Figure 5 sensors-23-06487-f005:**
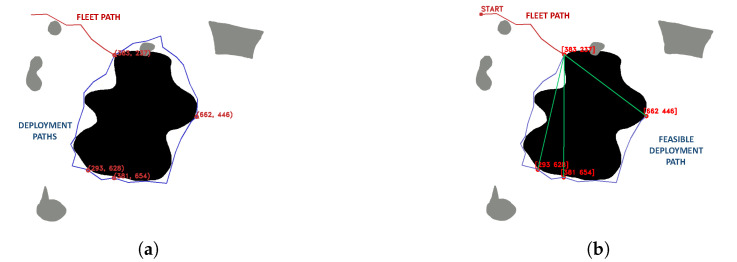
Deployment procedure, (**a**) definition of fleet path (red) in node-edge configuration and deployment path (blue) in individual configuration, (**b**) feasible deployment configuration with roped restrictions. Source: Authors.

**Figure 6 sensors-23-06487-f006:**
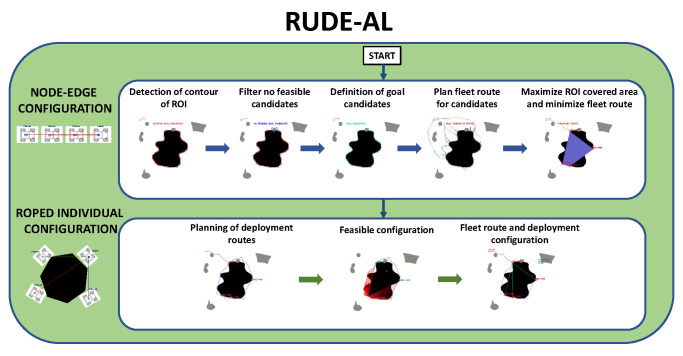
RUDE-AL algorithm procedure. Source: Authors.

**Figure 7 sensors-23-06487-f007:**
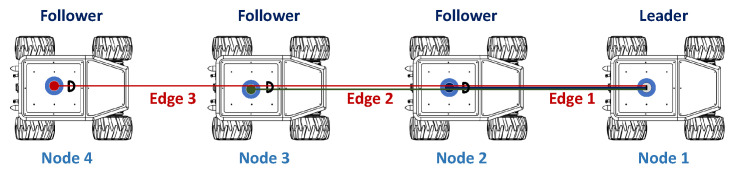
Node-edge configuration with cable robot on the leader robot. Source: Authors.

**Figure 8 sensors-23-06487-f008:**
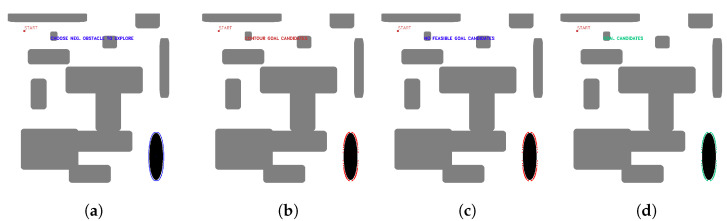
First stage candidate points with no nearby positive obstacles, (**a**) obstacle selection, (**b**) contour candidates, (**c**) evaluation of feasible and non-feasible candidates (red feasible, blue no feasible, (**d**) first stage candidate points (green). Source: Authors.

**Figure 9 sensors-23-06487-f009:**
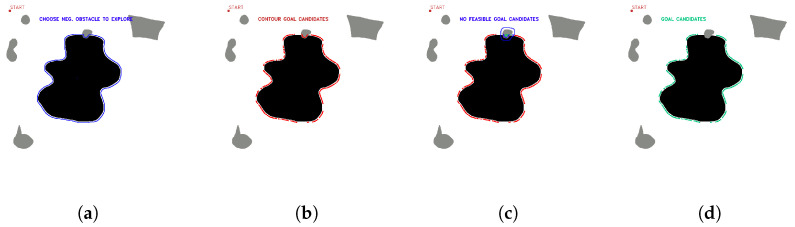
First stage candidate points with one nearby positive obstacles, (**a**) obstacle selection, (**b**) contour candidates, (**c**) evaluation of feasible and non-feasible candidates (red feasible, blue no feasible, (**d**) first stage candidate points (green). Source: Authors.

**Figure 10 sensors-23-06487-f010:**
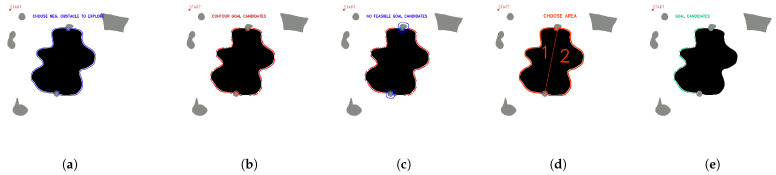
First stage candidate points with two nearby positive obstacles, (**a**) obstacle selection, (**b**) contour candidates, (**c**) evaluation of feasible and non-feasible candidates (red feasible, blue no feasible), (**d**) selection of available areas to explore, (**e**) first stage candidate points (green). Source: Authors.

**Figure 11 sensors-23-06487-f011:**
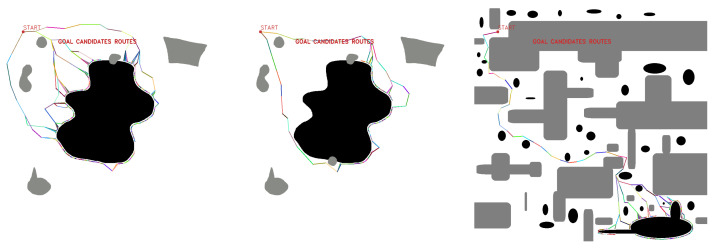
Multiple goal path planning for different scenarios. Source: Authors.

**Figure 12 sensors-23-06487-f012:**
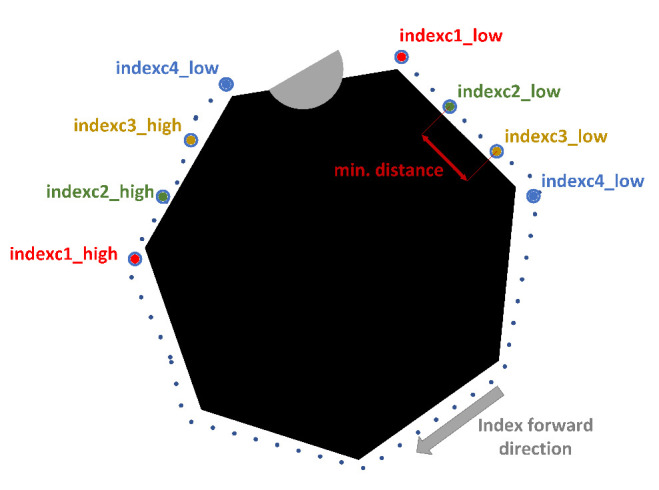
Definition of minimum and maximum index for each candidate point. Source: Authors.

**Figure 13 sensors-23-06487-f013:**
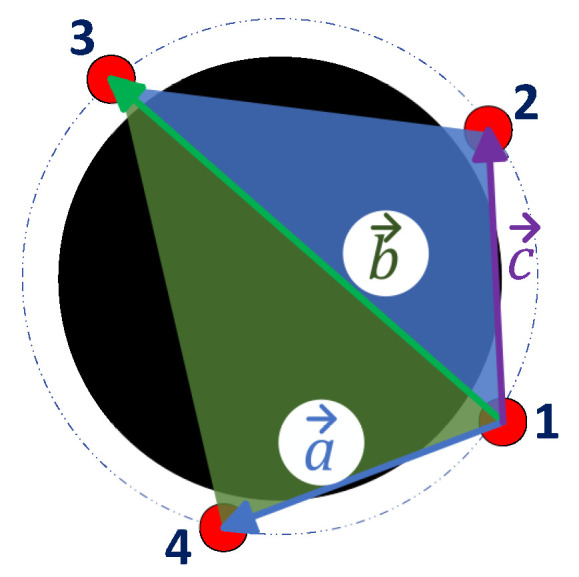
Vectors to calculate covered area. Source: Authors.

**Figure 14 sensors-23-06487-f014:**
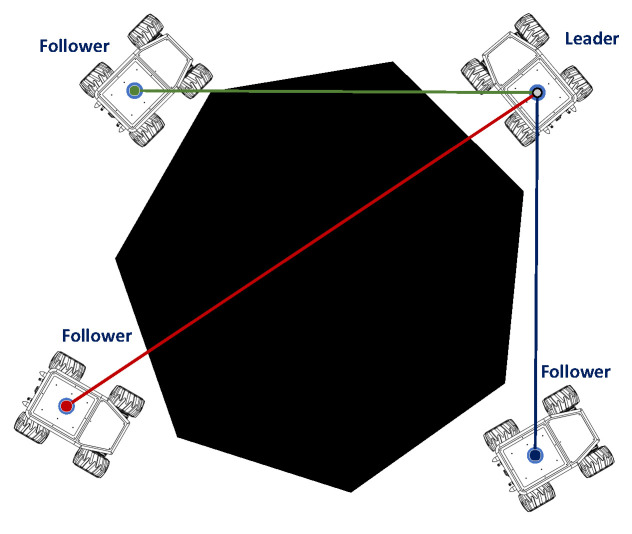
Roped individual configuration. Source: Authors.

**Figure 15 sensors-23-06487-f015:**
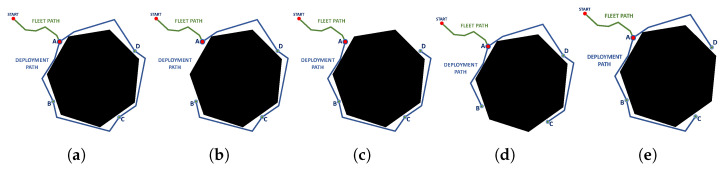
Candidate deployment configurations, (**a**) available deployment paths, (**b**) deployment path without A-B route, (**c**) deployment path without A-D route, (**d**) deployment path without B-C route, (**e**) deployment path without C-D route. Source: Authors.

**Figure 16 sensors-23-06487-f016:**
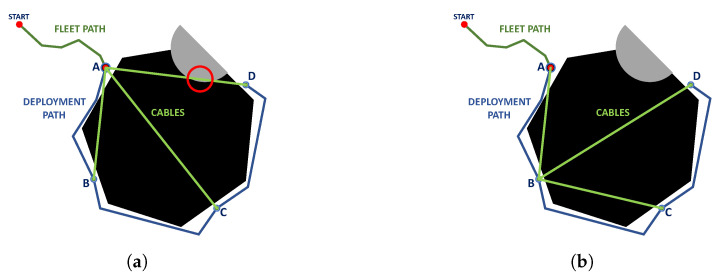
Cable robot position, (**a**) At point A (no roped feasible configuration because of collision), (**b**) At point B (roped feasible configuration). Source: Authors.

**Figure 17 sensors-23-06487-f017:**
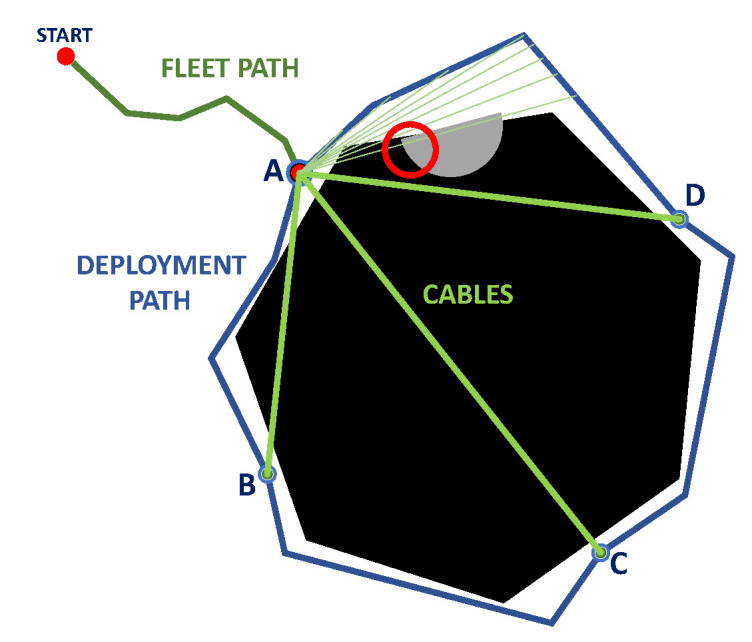
Evaluation of rope contact with positive obstacles at interpolated path for follower robot at A-D candidate route. Source: Authors.

**Figure 18 sensors-23-06487-f018:**
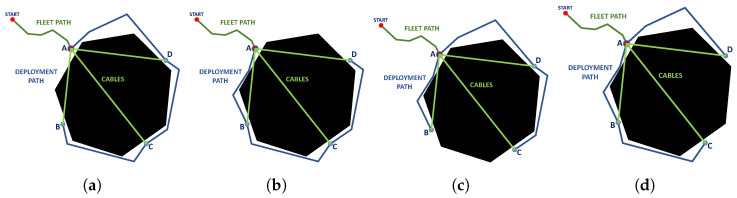
Feasible deployment path configuration using point A as principal node. (**a**) Type 1. (**b**) Type 2. (**c**) Type 3. (**d**) Type 4. Source: Authors.

**Figure 19 sensors-23-06487-f019:**
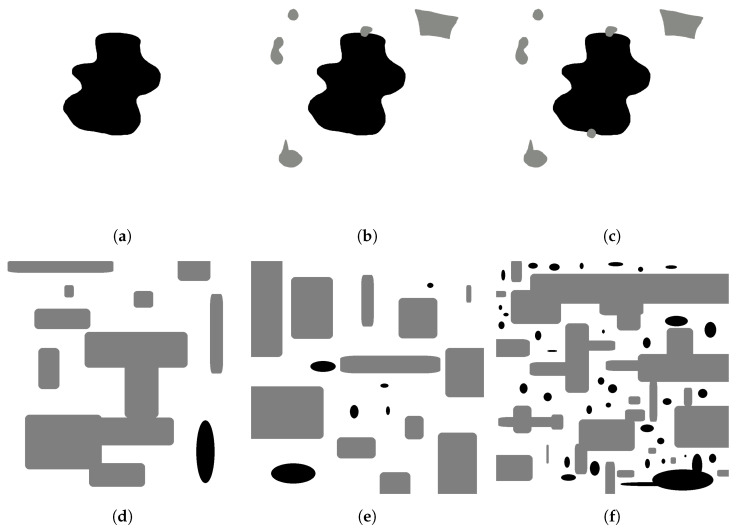
Manually generated maps for GLOBAL TEST, (**a**) map 1, (**b**) map 2, (**c**) map 3, (**d**) map 4, (**e**) map 5, (**f**) map 6. Source: Authors.

**Figure 20 sensors-23-06487-f020:**
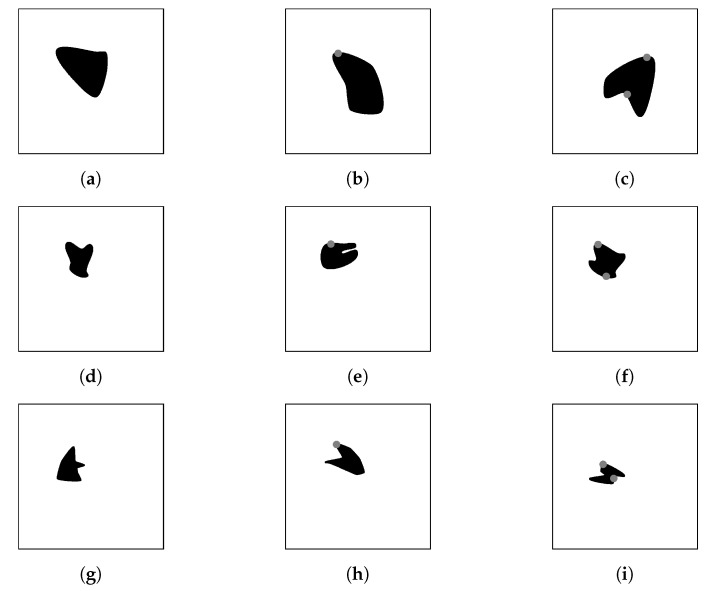
Automatic generated maps for SHAPE TEST, (**a**) maps 1–10 with 0 nearby positive obstacles, (**b**) maps 1–10 with 1 nearby positive obstacles, (**c**) maps 1-10 with 2 nearby positive obstacles, (**d**) maps 11–20 with 0 nearby positive obstacles, (**e**) maps 11–20 with 1 nearby positive obstacles, (**f**) maps 11–20 with 2 nearby positive obstacles, (**g**) maps 21–30 with 0 nearby positive obstacles, (**h**) maps 21–30 with 1 nearby positive obstacles, (**i**) maps 21–30 with 2 nearby positive obstacles. Source: Authors.

**Figure 21 sensors-23-06487-f021:**
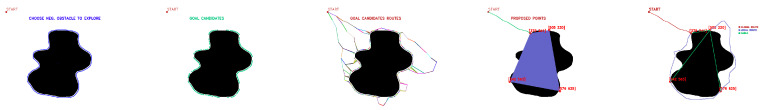
GLOBAL TEST procedure for Map 1. Source: Authors.

**Figure 22 sensors-23-06487-f022:**
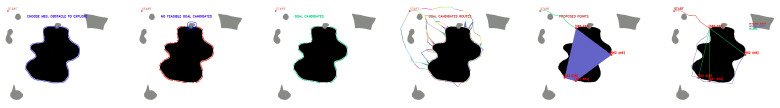
GLOBAL TEST procedure for Map 2. Source: Authors.

**Figure 23 sensors-23-06487-f023:**
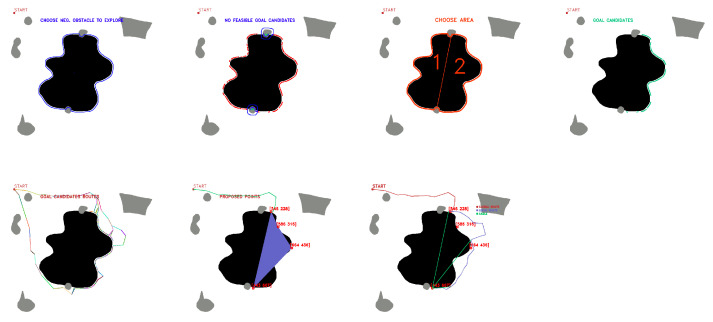
GLOBAL TEST procedure for Map 3. Source: Authors.

**Figure 24 sensors-23-06487-f024:**
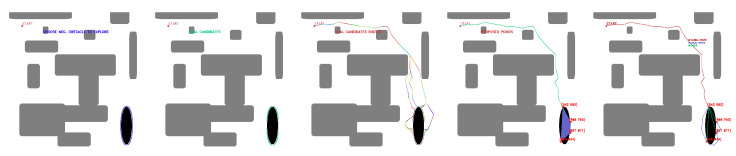
GLOBAL TEST procedure for Map 4. Source: Authors.

**Figure 25 sensors-23-06487-f025:**
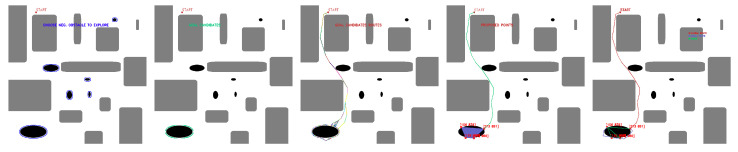
GLOBAL TEST procedure for Map 5. Source: Authors.

**Figure 26 sensors-23-06487-f026:**
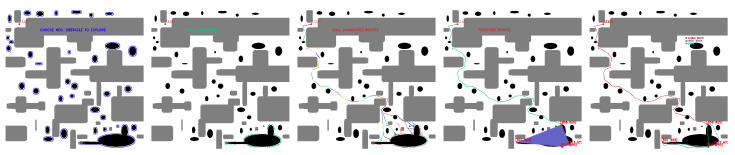
GLOBAL TEST procedure for Map 6. Source: Authors.

**Figure 27 sensors-23-06487-f027:**
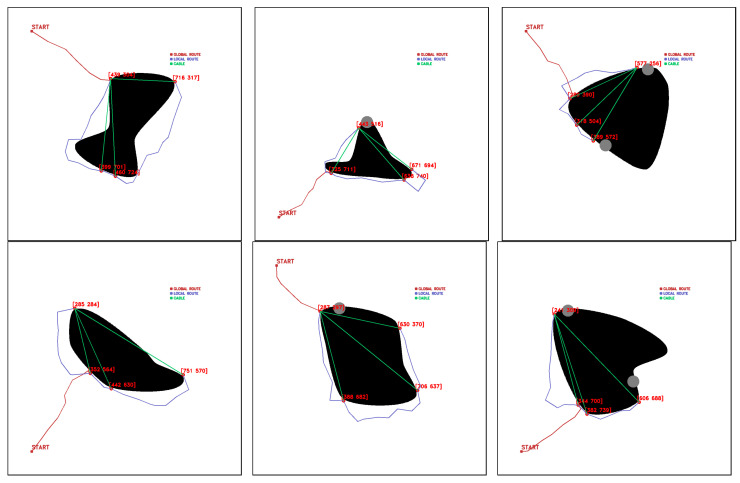
Feasible fleet and deployment paths tested in SHAPE TEST generated environment. Source: Authors.

**Figure 28 sensors-23-06487-f028:**
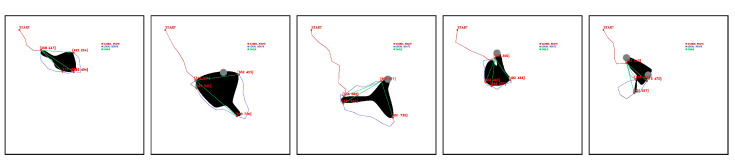
Selected points for feasible fleet and deployment paths that limit the release of the CDPR. Source: Authors.

**Figure 29 sensors-23-06487-f029:**
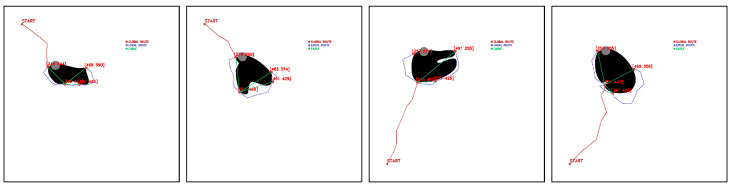
Selected points for feasible fleet and deployment paths close to hit positive obstacles during release of cable drive robot. Source: Authors.

**Figure 30 sensors-23-06487-f030:**
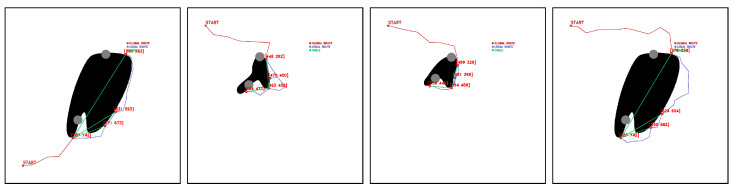
Selected feasible points that present problems during the release of the cable driven robot. Source: Authors.

**Figure 31 sensors-23-06487-f031:**
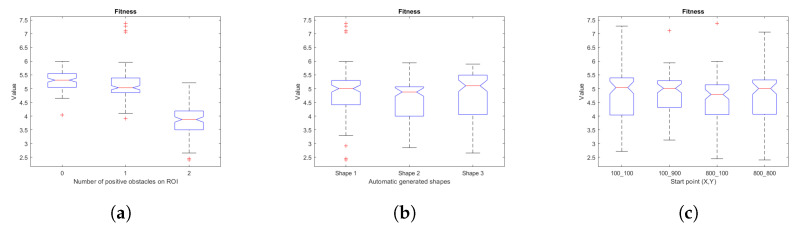
Box and whiskers diagrams for fitness and its variables, (**a**) fitness vs. number of positive obstacles on ROI, (**b**) fitness vs. automatic generated shapes, (**c**) fitness vs. start points. Source: Authors.

**Figure 32 sensors-23-06487-f032:**
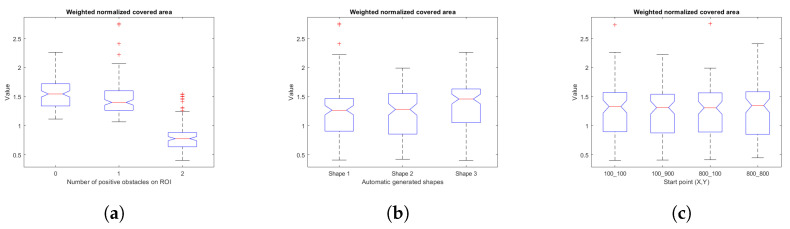
Box and whiskers diagrams for weighted normalized covered area and its variables, (**a**) weighted normalized covered area vs. number of positive obstacles on ROI, (**b**) weighted normalized covered area vs. automatic generated shapes, (**c**) weighted normalized covered area vs. start points. Source: Authors.

**Figure 33 sensors-23-06487-f033:**
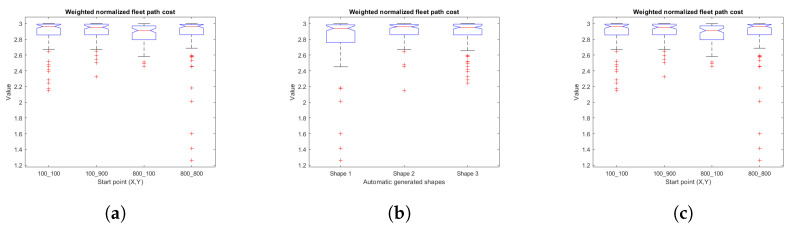
Box and whiskers diagrams for weighted normalized fleet path cost and its variables, (**a**) weighted normalized fleet path cost vs. number of positive obstacles on ROI, (**b**) weighted normalized fleet path cost vs. automatic generated shapes, (**c**) weighted normalized fleet path cost vs. start points. Source: Authors.

**Figure 34 sensors-23-06487-f034:**
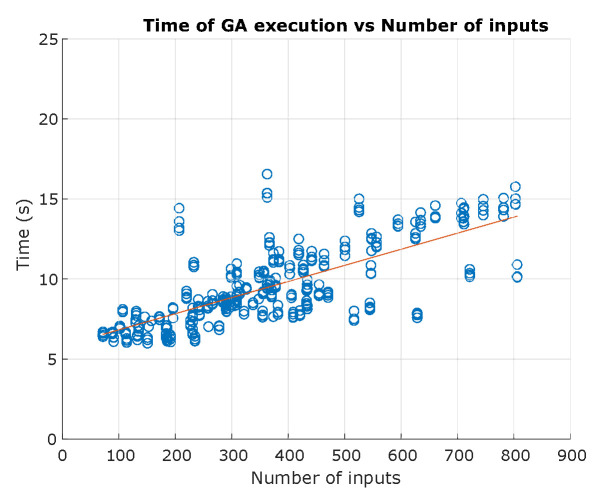
Execution time of GA algorithm. Source: Authors.

**Figure 35 sensors-23-06487-f035:**
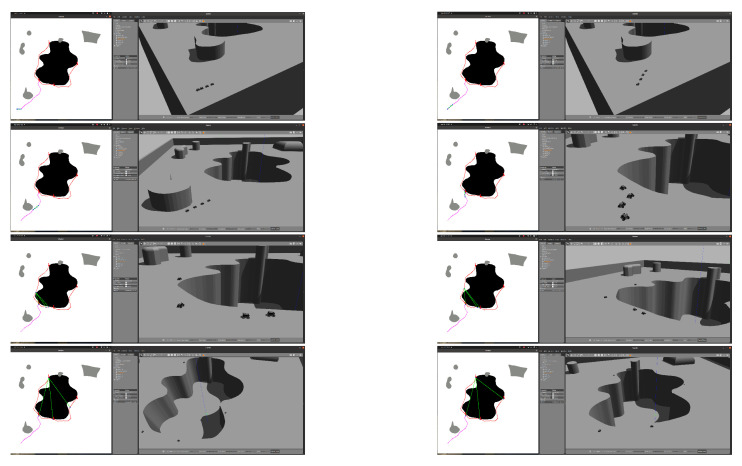
Gazebo simulation of navigation and deployment of the robot fleet around a sinkhole. Source: Authors.

**Table 1 sensors-23-06487-t001:** Comparison of state-of-the-art CDPR prototypes.

Reference	Robot Name	Mobile Bases	Application	Environment	Physical Implementation	Number of Robotic Platforms	Degrees of Freedom
[25]	Robocrane	YES	Research	Indoor/Outdoor	YES	3	6
[26]	IPAnema	NO	Industry	Indoor	YES	1	6
[27]	CoGiro	NO	Research	Indoor	YES	1	6
[28]	CableRobot	NO	Research	Indoor	YES	1	6
[29]	-	NO	Research	Indoor	YES	1	6
[30]	CABLAR	NO	Logistics	Indoor	YES	1	6
[31]	FASTKIT	YES	Logistics	Indoor	YES	2	6
[32]	MARIONET-CRANE	YES	Rescue	Outdoor	YES	1	6
[33]	MoPick	YES	Logistics	Indoor	YES	4	3
[34]	-	YES	Research	Outdoor	NO	4	4
[35]	-	YES	Research	Outdoor	NO	3	3
[36]	-	YES	Research	Outdoor	NO	3	3

**Table 2 sensors-23-06487-t002:** RGB color definition used in 2D maps for RUDE-AL. Source: Authors.

Tipo	RGB Color	UGVs Navigation Allowed	Cable Cross Allowed
Free space	(255, 255, 255)	YES	YES
Positive obstacle	(136, 138, 133)	NO	NO
Negative obstacle	(0, 0, 0)	NO	YES

**Table 3 sensors-23-06487-t003:** Parameters defined in PyGAD instance.

Parameter	Definition	Value
num_generations	Number of generations	1200
mutation_num_genes	Number of genes por instance random_mutation()	4
num_parents_mating	Number of solutions to be selected as parents	2
sol_per_pop	Number of solutions in the population	70
num_genes	Number of genes in each solution	4
fitness_function	Fitness function	$f_{fitness}$
init_range_low	Lower value of random range where initial population is selected.	0
init_range_high	Upper value of random range where initial population is selected.	len(candidate1)/2
crossover_type	Type of crossover operation.	“single_point”
random_mutation_min_val	Start value of the range from which a random value is selected to be added to the gene.	1
random_mutation_max_val	End value of the range from which a random value is selected to be added to the gene.	100
mutation_type	Type of mutation operation	“random”
gene_space	Specify the posible values for each gene in order to restrict the gene values.	[range (0, len(candidate1)), range (0, len(candidate2)), range (0, len(candidate3)), range (0, len(candidate4)) ]
gene_type	Gene type (numeric data type)	int

**Table 4 sensors-23-06487-t004:** Feasible configuration for deployment path.

Point	1st Feasible Path	2nd Feasible Path	3rd Feasible Path	Feasible Configuration
A	$R_{1}$	$R_{2}$	$R_{3}$	A →($R_{1}$,$R_{2}$,$R_{3}$)
	$R_{1}$	$R_{2}$	$R_{4}$	A →($R_{1}$,$R_{2}$,$R_{4}$)
	$R_{1}$	$R_{3}$	$R_{4}$	A →($R_{1}$,$R_{3}$,$R_{4}$)
	$R_{1}$	$R_{3}$	$R_{4}$	A →($R_{2}$,$R_{3}$,$R_{4}$)
B	$R_{1}$	$R_{2}$	$R_{3}$	B →($R_{1}$,$R_{2}$,$R_{3}$)
	$R_{1}$	$R_{2}$	$R_{4}$	B →($R_{1}$,$R_{2}$,$R_{4}$)
	$R_{1}$	$R_{3}$	$R_{4}$	B →($R_{1}$,$R_{3}$,$R_{4}$)
	$R_{1}$	$R_{3}$	$R_{4}$	B →($R_{2}$,$R_{3}$,$R_{4}$)
C	$R_{1}$	$R_{2}$	$R_{3}$	C →($R_{1}$,$R_{2}$,$R_{3}$)
	$R_{1}$	$R_{2}$	$R_{4}$	C →($R_{1}$,$R_{2}$,$R_{4}$)
	$R_{1}$	$R_{3}$	$R_{4}$	C →($R_{1}$,$R_{3}$,$R_{4}$)
	$R_{1}$	$R_{3}$	$R_{4}$	C →($R_{2}$,$R_{3}$,$R_{4}$)
D	$R_{1}$	$R_{2}$	$R_{3}$	D →($R_{1}$,$R_{2}$,$R_{3}$)
	$R_{1}$	$R_{2}$	$R_{4}$	D →($R_{1}$,$R_{2}$,$R_{4}$)
	$R_{1}$	$R_{3}$	$R_{4}$	D →($R_{1}$,$R_{3}$,$R_{4}$)
	$R_{1}$	$R_{3}$	$R_{4}$	D →($R_{2}$,$R_{3}$,$R_{4}$)

**Table 5 sensors-23-06487-t005:** Definition of individual robot paths for feasible deployment path configurations.

Principal Node Point	Feasible Configuration	Robot 1 Path	Robot 2 Path	Robot 3 Path	Robot 4 Path
A	type 1	FP+ADCB	FP + ADC	FP + AD	FP
	type 2	FP + ABCD	FP + ABC	FP + AB	FP
	type 3	FP + ADC	FP + AD	FP + AB	FP
	type 4	FP + ABC	FP + AB	FP + AD	FP
B	type 1	FP + BADC	FP + BAD	FP + BA	FP
	type 2	FP + BCDA	FP + BCD	FP + BC	FP
	type 3	FP + BAD	FP + BA	FP + BC	FP
	type 4	FP + BCD	FP + BC	FP + BA	FP
C	type 1	FP + CBAD	FP + CBA	FP + CB	FP
	type 2	FP + CDAB	FP + CDA	FP + CD	FP
	type 3	FP + CBA	FP + CB	FP + CD	FP
	type 4	FP + CDA	FP + CD	FP + CB	FP
D	type 1	FP + DCBA	FP + DCB	FP + DC	FP
	type 2	FP + DABC	FP + DAB	FP + DA	FP
	type 3	FP + DCB	FP + DC	FP + DA	FP
	type 4	FP + DAB	FP + DA	FP + DC	FP

**Table 6 sensors-23-06487-t006:** Characteristics of GLOBAL TEST environment maps.

Map	Number of Positive Obstacles	Number of Negative Obstacles	Number of Positive Obstacles around ROI	Shape of Negative Obstacle
1	0	1	0	Irregular
2	4	1	1	Irregular
3	5	1	2	Irregular
4	8	1	0	Ellipse
5	12	6	0	Ellipse
6	17	34	0	Irregular ellipse

**Table 7 sensors-23-06487-t007:** Characteristics for automatic generated maps.

Maps	Number of Positive Obstacles around ROI	${\mathrm{Cp}}_{\mathrm{rad}}$	Smoothness	$N_{\mathrm{random}}$	Scale
1–10	0–2	0.2	0.05	6	500
11–20	0–2	0.3	0.08	7	250
21–30	0–2	0.1	0.1	8	300

**Table 8 sensors-23-06487-t008:** Summary of results.

Experiment	Number of Maps	Number of Experiments	Number of Successful Experiments	Successful Rate
Global TEST	6	24	24	100%
SHAPE TEST (feasible paths)	90	360	357	99.16%
SHAPE TEST (feasible paths with workspace limitation)	90	360	352	97.78%
SHAPE TEST (feasible paths with collision risk)	90	360	350	97.2%
SHAPE TEST (feasible paths with exceptions)	90	360	354	98.33%
SHAPE TEST	90	360	333	92.5%

## Data Availability

A video description of the article can be found on Appendix A. The reported data can be found on Appendix B.

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
