# Peer review of "RUDE-AL: Roped UGV Deployment Algorithm of an MCDPR for Sinkhole Exploration"

_sensors, 2023, doi:10.3390/s23146487_

Round 1

Reviewer 1 Report

1.       The main objective of the research is the deployment of a Mobile Cable Driven Parallel Robot inside of a sinkhole, using different mobile robots. The researchers developed an algorithm, that can select target points for the mobile robots, considering the constraints of cable driven fleet, to minimize the cost of the robot’s deployment distance and maximize the area covered for sinkhole exploration tasks.

2.       The article is well structured as follows: Section 2 presents the state-of-the-art and related works which are well founded. Section 3 details the problem formulation, the proposal and the description of the experiments performed. Section 4 shows the results obtained by the target point selection algorithm. Finally, Section 5 presents the conclusions obtained.

3.       The research can be considered both original and relevant to the field. The deployment of a Mobile Cable Driven Parallel Robot (MCDPR) inside a sinkhole using a fleet of mobile robots is a new approach in the field of exploratory robotics. The concept can be considered a gap in the field by presenting a solution for sinkhole exploration, combining locomotion of mobile robots and a cable-driven parallel robot. This integrated approach fills a gap in the existing methods for robotic exploration and offers potential advancements in the field of robotics and disaster response.

4.       The conclusions drawn from the research are consistent with the evidence and arguments presented throughout the study. The researchers have meticulously analyzed and interpreted the data collected from the experiments, observations, and simulations conducted. They have effectively supported their claims with logical reasoning, empirical evidence, and theoretical foundations.

5.       Furthermore, the conclusions directly address the main question posed in the research. The study aims to investigate the feasibility and effectiveness of deploying a Mobile Cable Driven Parallel Robot (MCDPR) inside a sinkhole using a fleet of mobile robots. The researchers have provided a comprehensive analysis of the results, demonstrating the capabilities and limitations of the proposed approach. They have effectively addressed the main question by presenting the findings that validate the potential of using MCDPRs and mobile robots for sinkhole exploration and assistance.

6.       In summary, the conclusions are well-aligned with the evidence and arguments presented, and they successfully address the main question posed in the research.

7.       The paper can be published in a journal like SENSORS.

Reviewer 2 Report

In this paper, the authors consider RUDE-AL (Roped UGV Deployment Algorithm), a methodology for deploying a Mobile Cable Driven Parallel Robot (MCDPR) composed of four mobile robots and a cable driven parallel robot (CDPR) for sinkhole exploration tasks and assistance to potential trapped victims. The deployment of the fleet is organized with node-edge formation during the mission’s first stage, positioning itself around the area of interest and acting as anchors for the subsequent release of the cable robot. The topic is interesting. I have the following comments for the further improvement of the paper.

1) There is a lot of relevant work in the paper, and it is suggested that the author add a table to sort and compare it.

2) The author is suggested to summarize the contribution points of the paper.

3) The authors can discuss the time and space complexity of the algorithm and compare it with other bionic evolutionary algorithms such as those in Path planning and energy efficiency of heterogeneous mobile robots using cuckoo–beetle swarm search algorithms with applications in UGV obstacle avoidance; Path planning and smoothing of mobile robot based on improved artificial fish swarm algorithm

4) The authors are suggested to present more future potential works in the conclusion part.

The paper is good writing and presents technical contributions, which could be accepted after a revision.

 Minor editing of English language required.

Author Response

Dear Reviewer,

Special Issue

"Mobile Robots: Navigation, Control and Sensing"

Subject: Submission of revised article 1st round to "Sensors"

Dear Reviewer,

We are pleased to send the modified version of our article “RUDE-AL: Roped UGV deployment algorithm of a MCDPR for sinkhole exploration”. We appreciate your positive assessment of our article and we value your effort and time spent reviewing it.

We have addressed all the points suggested by the reviewer, especially those referring to comparisons of the state-of-the-art methods, new subsections for algorithm complexity analysis and future works. We hope that after these reviews the article will be approved for publication, at the end of the document you can find the changes made highlighted.

Sincerely.

David Orbea, Christyan Cruz Ulloa, Jaime del Cerro and Antonio Barrientos*.

Center for Automation and Robotics – Robotics and Cybernetics Research Group

Universidad Politécnica de Madrid C/ José Gutiérrez Abascal, 2.

28006 Madrid

Tlf: +34 604125284

Spain

https://www.car.upm-csic.es/

Reviewer 2

Point 1: There is a lot of relevant work in the paper, and it is suggested that the author add a table to sort and compare it.

Response: Thanks for the suggestion to improve our work. A table has been added in the Section 2. Related Work, comparing the main characteristics of state-of-the-art works according to the capabilities of the developed prototypes for specific tasks, environment, number of robots, degrees of freedom, use of mobile bases (for MCDPR).

Point 2: The author is suggested to summarize the contribution points of the paper.

Response:. The reviewer’s suggestion is relevant. In Section 1. Introduction, the main contribution points have been summarized, specifying that the goal of the presented approach is the autonomous selection of anchoring points of a mobile robot fleet. Also, redaction has been improved to help the reader to understand the problem solution approach covered in this article.

Point 3: The authors can discuss the time and space complexity of the algorithm and compare it with other bionic evolutionary algorithms such as those in Path planning and energy efficiency of heterogeneous mobile robots using cuckoo–beetle swarm search algorithms with applications in UGV obstacle avoidance; Path planning and smoothing of mobile robot based on improved artificial fish swarm algorithm.

Response: The reviewer’s observation is appreciated. The algorithm complexity analysis has been added to the paper in Section 5.2. Algorithm Complexity. The time analysis complexity analysis has been realized considering the execution time of the evolutionary algorithm, obtaining a fair complexity result for the input range (candidate points to optimize according to the fitness function), concluding that the behavior of the algorithm is good for our purposes. As the main goal of the paper is the autonomous selection of feasible anchoring points, our results and analysis are focused on traversability, feasible paths for the mobile fleet with the roped restrictions, and feasible configuration to deploy the robotic fleet for exploration of sinkholes.

Point 4: The authors are suggested to present more future potential works in the conclusion part.

Response: Thanks for the suggestion to improve our work. In Section 6. Conclusions, more potential future works has been added related to the navigation, simulation, dynamical restrictions and the workspace of the CDPR that will be deployed in the sinkhole.

Round 2

Reviewer 2 Report

I have no comment.

I have no comment.